# REIP: A Reconfigurable Environmental Intelligence Platform and Software Framework for Fast Sensor Network Prototyping

**DOI:** 10.3390/s22103809

**Published:** 2022-05-17

**Authors:** Yurii Piadyk, Bea Steers, Charlie Mydlarz, Mahin Salman, Magdalena Fuentes, Junaid Khan, Hong Jiang, Kaan Ozbay, Juan Pablo Bello, Claudio Silva

**Affiliations:** Tandon School of Engineering, New York University, Brooklyn, NY 11201, USA; bsteers@nyu.edu (B.S.); cmydlarz@nyu.edu (C.M.); ms6617@nyu.edu (M.S.); mfuentes@nyu.edu (M.F.); khanj@wwu.edu (J.K.); hj1274@nyu.edu (H.J.); kaan.ozbay@nyu.edu (K.O.); jpbello@nyu.edu (J.P.B.); csilva@nyu.edu (C.S.)

**Keywords:** heterogeneous sensor networks, open-source, multi-modal, Internet of Things (IoT), SDK, modular API, Python, multiprocessing

## Abstract

Sensor networks have dynamically expanded our ability to monitor and study the world. Their presence and need keep increasing, and new hardware configurations expand the range of physical stimuli that can be accurately recorded. Sensors are also no longer simply recording the data, they process it and transform into something useful before uploading to the cloud. However, building sensor networks is costly and very time consuming. It is difficult to build upon other people’s work and there are only a few open-source solutions for integrating different devices and sensing modalities. We introduce REIP, a Reconfigurable Environmental Intelligence Platform for fast sensor network prototyping. REIP’s first and most central tool, implemented in this work, is an open-source software framework, an SDK, with a flexible modular API for data collection and analysis using multiple sensing modalities. REIP is developed with the aim of being user-friendly, device-agnostic, and easily extensible, allowing for fast prototyping of heterogeneous sensor networks. Furthermore, our software framework is implemented in Python to reduce the entrance barrier for future contributions. We demonstrate the potential and versatility of REIP in real world applications, along with performance studies and benchmark REIP SDK against similar systems.

## 1. Introduction

Sensor networks have expanded our ability to monitor and study the world. They have been used for a wide range of applications, such as monitoring air pollution [1], urban noise [2] or energy management of smart buildings [3]. As their use cases expand, sensor networks become more complex and powerful, enabling a new range of physical stimuli to be accurately recorded, processed, ingested and analysed. However, implementing sensor networks is an enormous endeavour with high cost in time and resources. Many decisions have to be made, from which devices to use to which protocols to employ for connecting them. In addition, the deployment and upkeep of sensor networks is critical and time consuming, which requires sophisticated monitoring and alerting tools. Nowadays, it is difficult to build on top of other people’s work as there are few accessible open-source solutions suitable for integration into different devices, leading to countless hours of engineering and software design invested every time.

Of note is the case of high throughput sensor applications that incorporate audio or video data capture. Multithreading is typically needed to enable concurrent data capture, processing and writing to disk. If not handled correctly multiple threads accessing hardware devices or disk locations can lead to race conditions that can result in data corruption or even hardware freezes. Race conditions and hardware lockups can be incredibly difficult to identify and diagnose and are usually only addressed by more experienced developers rather than domain specific researchers implementing sensors networks. Sensor networks deployed externally in hard to reach locations have a critical need for stability over long periods of time and are particularly sensitive to these kinds of thread-borne failures, which can manifest at arbitrary times, often after many hours, weeks or even months of operation. The cost of addressing these failures in the field is high; thus, ensuring stable code operation is key.

REIP is a Reconfigurable Environmental Intelligence Platform for fast sensor network prototyping, including an efficient and scalable sensor runtime. Given a sensing application and a set of requirements, REIP aims to alleviate the work of designing a remote sensor network by providing tools for sensor node design, software and hardware integration, bandwidth management, and other time-costly aspects.

In this work, we present the first and central tool of REIP referred to as the REIP SDK (or software framework)—a modular API containing a set of re-usable, plug-and-play software blocks that integrate solutions for different hardware and software components by following the best practices. Moreover, we implement this API as a flexible, open-source software framework in Python, with the goal of it being user-friendly and, most importantly, to encourage contributions and extensions in the future. Its block libraries can also serve as a reference on how to implement different aspects of the sensor software even when users choose/need to build a custom solution. (The repository is hosted at https://github.com/reip-project/reip-pipelines (updated on 25 April 2022)).

It should be noted that our work is aimed at building sensors with edge computing capabilities of Linux-based high performance SBCs (Single Board Computers), such as the NVIDIA Jetson [4] or Raspberry Pi [5,6]. The amount of data generated by modern sensing platforms (especially those containing video cameras) is such that it is often infeasible to upload all of it to the cloud for later processing. On-the-edge real time processing of sensor data is required to filter out the background noise or generate more compact representations of the data, and in such scenarios an efficient utilisation of the hardware capabilities offered by the computing platform is key. REIP SDK was designed to have minimal performance overhead on such platforms and offers the user full control over the execution of data acquisition of the processing pipelines.

The contributions of this work are listed in the following:A concept of the REIP platform for fast prototyping of heterogeneous sensor networks;An open-source implementation of the REIP SDK for rapid development of multimodal sensors with edge computing capabilities;Performance evaluation of the REIP SDK  under different configurations, including comparison with other existing software frameworks;Extensive benchmarking results from different hardware platforms demonstrating minimal overhead and scalability of the REIP SDK;A case study highlighting the utility of the REIP SDK in designing multimodal sensors.

The structure of the paper is as follows: in Section 2, we discuss how REIP as a platform and the REIP SDK stand in relation to existing solutions; in Section 3, we describe the design approach and API of the REIP SDK; in Section 4, we present performance studies regarding concurrency and overhead, and report the performance of the REIP SDK on different hardware platforms. Furthermore, we contrast the performance of our SDK to similar solutions for a representative use case in Section 4.1.5. In Section 5, we present an example of using REIP in a real world setting and discuss the potential for the platform’s use in multi-modal sensing applications. Finally, we provide concluding remarks and a vision of future components of REIP (SDK) in Section 6.

## 2. Related Work

Sensor networks are being used in a large range of applications, each with varying computing requirements and ranging from sensing a single modality with low data loads (e.g., intermittent air quality sensing [7]) to more complex and heterogeneous sensor networks with larger data flows and computing requirements (e.g., audio-visual traffic monitoring [8] or sports analytics [9]). Existing frameworks for sensor network development are typically designed to work on a narrow range of requirements, e.g., low data volumes [10] or large computing resources [11]. As sensor networks become more common, and expand their sensing modalities, along with the applications they serve, these frameworks fall short in terms of flexibility and re-usability across different hardware platforms. Table 1 provides a feature comparison of different existing sensor network development platforms (top half) as well as software frameworks for building data acquisition and processing pipelines (bottom half). We emphasise that all these projects have their own, sometimes opposite, design goals and can often be complementary to what is proposed in our work. Nonetheless, we try to assess every project on all seven criteria to provide as complete an overview as possible. When the framework can not be evaluated directly, because it depends on custom hardware or is not open source, we rely on the results reported in the corresponding publications materials. In the following section, we describe the most relevant sensor network platforms, what they excel at, and how they are different from REIP (top half of Table 1).

### 2.1. Sensor Network Platforms

Solutions for the sustainable and reusable development of sensor networks have been explored before in the context of industry as well as in academia. Different alternatives have been proposed, which tackle common challenges such as hardware/software (HW/SW) integration and/or the use of heterogeneous devices, being open-source with user-friendly API, etc.

Among notable platforms with scalable HW/SW integration is FIT-IoT Lab [12], which is a test-bed available for researchers addressing wireless communications in sensor networks and low power routing protocols, with embedded and distributed applications. However, it is not device-agnostic, as only a limited set of sensors are supported with no extensibility. Similarly, FIESTA IoT [13] is a meta-testbed IoT/cloud infrastructure designed for the submission of experiments over interconnected hardware testbeds using a single set of credentials. Although it is scalable, it still lacks an extensible and open-source API and thus relies on proprietary testbed deployments associated with the institutions in the FIESTA IoT Consortium.

Other solutions such as Signpost [14], which is an extensible solar energy-harvesting modular platform designed to enable city-scale deployments, are not device-agnostic and only work with the customised sensors they provide. It also requires a considerable amount of engineering to reproduce a sensor network in this configuration. In contrast, SensorCentral [15] is a device-agnostic, multimodal sensor platform but it does not consider HW/SW integration at the sensor level and is not open-source for the research community to use.

The most similar to the REIP platform is the Array of Things (AoT) [16,17], an urban sensing system designed to collect real-time data from the environment leveraging a sensor platform called Waggle [24]. The AoT comprises an open-source API with multiple sensing modalities; however, it does not meet the device-agnostic criteria, as it depends heavily on the Waggle platform which itself cannot be easily produced at scale by other institutions/researchers.

Similarly, WaspMote [18] offers a modular hardware and software architecture integration, with an open-source API to create its application pipelines. Some popular use cases for the system include smart cities, water and agriculture applications, most of which utilise low bandwidth wireless technologies such as LoRa or ZigBee. WaspMote was designed for low volume IoT applications on constrained and generally battery powered edge devices such as Micro Controller Units (MCUs), which are not typically suited for processing high volumes of audio or video data [32]. REIP, in contrast, is targeting higher performance Linux-based SBCs to tackle such data intensive sensing applications.

The USC test-bed [19] currently under construction is to be a scalable HW/SW integrated sensor network with sensors, actuators and wireless radios to support experimental research on sensing, processing, algorithms and software for IoT. However, an open-source codebase and user-friendly API are so far not announced as part of their design goals.

Ultimately, the aim of REIP is to provide a sensor network development platform that meets all of the before-mentioned criteria in a balanced way. As the first step, we implement and present in this work the REIP SDK—an open-source device-agnostic SDK/API, that supports multiple sensing modalities and hardware/software integration, and which is not only user-friendly but is also scalable and extensible. The following section details existing software frameworks that are relevant to the creation of data acquisition/processing *pipelines* such as the ones created using the REIP SDK for sensing platforms with edge computing capabilities (bottom half of Table 1).

### 2.2. Software Pipeline Frameworks

It is natural to develop sensor software as a data acquisition/processing *pipeline* since they typically contain a data source (e.g., a camera), some form of data processing and, often, a network layer (although some sensors are standalone devices that store data locally). There exists a variety of software frameworks for building pipelines in different application domains but none are out-of-the-box a good fit for building a software stack with real-time performance on sensors with edge computing capabilities, in particular, modern SBCs. Internet of Things (IoT) pipeline frameworks (e.g., FIWARE [20]) are mainly designed to work with multiple streams of small data packets, where the computational cost of data serialisation is not a major concern. Other big data frameworks, such as Apache Airflow [33] or Ray [26], are developed for handling large volumes of data across computing clusters, and are not suitable for running in real-time on IoT devices because of large performance overhead and non-trivial compilation steps on embedded systems. Multimedia pipeline frameworks such as GStreamer [29] are designed for highly-efficient multimedia applications, but their implementation and documentation are difficult to understand for anyone who is not an expert, and thus are time consuming or in some cases not practical to extend to more specialised sensing applications.

We elaborate on different existing software pipeline frameworks in the following:
IoT Frameworks

IoT software frameworks for building sensor runtimes are mainly designed in a light-weight manner, assuming small packets of data, e.g., sporadic IoT events or low volume data streams such as temperature measurements [34]. They do not scale well to larger data streams, such as video processing with machine learning on the edge [35].

FIWARE [20] is an open-source solution with a focus on smart city applications. It makes use of multiple programming languages, and the community provides docker images of various implementations with different run-time requirements. Because of this versatility, extending FIWARE for custom implementation requires full-stack development knowledge of these languages (e.g., Java, Node.js, C++ and Python), which implies a steep learning curve [36]. The limitations of FIWARE have been highlighted by performance evaluations of the platform [37]. Users state a high barrier to entry and the platform shows high performance with low volume event data but introduces latency when operating over larger scale wireless sensor networks. DDFlow [21], a visual and declarative programming abstraction, is a significant contribution to heterogenous IoT networks. Its goal is to provide a flexible programming framework without burdening users with low level hardware and network details, such as load balancing. The runtime interface utilises available resources to dynamically scale and map an IoT application similar to EdgeProg [22]. While DDFlow can be used for multi-modal sensing, the library is very high level and does not currently have the capabilities to facilitate high throughput application pipelines [38].

Other solutions, such as Caesar [23], are either not open-source and lack HW/SW integration, or are simply designed to handle small data loads as is the case with Waggle [25]. In addition, Caesar is limited to activity recognition using cameras as an application only and, while supporting multiple modalities, Waggle relies on the builtin parallelisation and data serialisation libraries in Python, which make it impractical to use for high data throughput applications. The REIP SDK, in contrast, offers multiple parallelisation and data serialisation strategies to best match the application needs (Section 4).

Big Data Frameworks

Distributed big data frameworks have gained traction and have been under heavy development in academia and industry for their ability to concurrently process large amounts of data. These frameworks, including Apache Ray [26], Spark [39], Celery [27] and Spotify’s Luigi [28], provide many of the concepts we are looking for with regards to the modular design of complex pipelines. However, the intended use-case of these frameworks is different in that it focuses on task scheduling, tracking, dependency resolution, and coordination across a cluster of machines. These lead to additional serialisation and increased latency that are unnecessary on a single local device, causing difficulty in scaling pipelines to fit constrained devices [40]. Ultimately, they were designed for use on the server-side and are much better suited for aggregating the data extracted from sensing platforms, rather than running on them.

The most comparable framework for our target use case is Apache Ray. Ray is a universal API for building distributed applications that enable end users to parallelise machine code across multiple CPUs and machines [41]. The foundational library that Ray is built on is Apache Arrow [42], a data management library focused on the fast movement and processing of large amounts of data, which includes their own highly efficient array serialisation formats [43]. Part of Arrow’s offering is a shared memory server called Plasma Store [44], which supports memory mapping on Unix-based devices to minimise the overhead of data serialisation in multiprocessing applications. While these tools are very useful for facilitating multi-tasking applications, Ray and other big data libraries were designed to run on larger computing clusters and are too heavy to scale to edge devices. REIP’s software framework capitalises on parts of these existing solutions (e.g., the Plasma Store) suitable for efficient sensor development while remaining lightweight and accessible for lower power embedding platforms.

Multimedia Frameworks

Highly popular multimedia libraries designed to handle large streams of video and audio data on a variety of devices are GStreamer [29], NVIDIA DeepStream [30] and FFmpeg [31]. GStreamer is an open-source pipeline/graph-based multimedia framework for complex data workflows, used in a variety of multimedia applications such as video editing, transcoding or streaming media. GStreamer is multi-platform and has been used reliably in pipelines for decades, but it is written in C and requires low-level programming expertise to extend it by implementing custom components outside of the scope of processing video/audio data types. These characteristics make it fall short to be a viable candidate for an easily-extensible and generalizable framework for the development of full application pipelines for sensor networks [45]. Similarly, FFmpeg that was designed for the processing of video and audio files in a CLI (Command Line Interface) does not offer an API for extending it to other modalities.

Some of the GStreamer limitations have been partially addressed by NVIDIA DeepStream, a scalable framework for building and deploying AI-powered video analytics on the edge. DeepStream provides ready-to-use AI components, such as object detection on video frames, but it follows GStreamer’s API; thus, new components cannot be efficiently implemented in a high level programming language, such as Python, but rather are only used through Python bindings. The framework still presents a steep barrier to entry for beginners and has limited flexibility to extend for applications other than audio-visual. Unlike GStreamer, NVIDIA DeepStream is not cross-platform and is dependent on NVIDIA platforms such as the Jetson family [5].

### 2.3. REIP SDK

Finally, the REIP SDK contributes a unique integration of data-flow programming abstractions and system implementation components that meet the productivity and performance needs of real-time IoT data collection and analysis applications. Many design choices in the REIP SDK were made in response to the challenges faced during the design of sensors and it is intended to accelerate the development of the software and integration of different components in a sensor network. We describe our design approach in the following section.

## 3. Approach

REIP seeks to provide a flexible and versatile environment for users to build, extend, reconfigure, and share their application code as they move through the rapid development process of designing sensor networks and other IoT data processing pipelines. In order to facilitate this, the REIP SDK provides a small number of abstractions so that users can take their existing code and integrate it seamlessly into a multi-tasking application.

### 3.1. The REIP SDK as Part of Sensor Network Development

A common workflow for sensor network prototyping using REIP is depicted in Figure 1. It contains the following typical steps:Define the project requirements, i.e., sensing modalities, sampling frequency, etc.Use the REIP SDK to build the data collection and processing pipeline. Custom blocks can be defined specific to the project needs, e.g., data processing with machine learning.Evaluate the data collection and processing pipeline and select the optimal edge compute platform.Implement any custom blocks and carry out the sensor build.Install the REIP SDK runtime on all of the edge sensors and the server.Deploy sensor network for data collection and processing.

In this paper, we focus on the REIP SDK, its API, performance and design principles. We discuss in Section 6 other components of the REIP platform that we foresee building in the future (i.e., a simulator to speed up step 3 in the workflow).

### 3.2. Benefits of the REIP SDK

When building application code handling data streaming and processing, it is often beneficial to decouple the program into smaller pieces that can run independently at their own rates using either multithreading and/or multiprocessing, which will prevent computationally demanding parts of the program from inhibiting the rest of the program (e.g., prevent machine learning from blocking video sampling). However, doing so often requires solving and scaffolding the same problems over and over again: How should the data be moved around? What serialisation method to use? What about error handling? and so on. There is a lot to consider when writing parallelised code, and when one has several applications, re-implementing solutions to the same problem while mixing the parallelisation logic with the application logic, it creates software that is very difficult to maintain and reuse in new projects.

The REIP SDK formalises design patterns that emerge repeatedly in many sensing applications. Typically, one has a collection of workers, each with some initialisation, data processing logic, and cleanup, where each of these workers communicates with others via thread-/process-safe queues for data sharing and management. Seemingly simple, such an approach can quickly result in a difficult to maintain code base for complex applications, resulting in deadlocks or other issues common to multitasking implementations. It takes a lot of effort and domain knowledge to structure such code properly. The REIP SDK offers a unique approach to making the implementation of data acquisition and processing software fast, easy and reliable in multi-worker contexts.

### 3.3. Design Principles

The REIP SDK seeks to cater to a wide range of domains, programmer expertise, and compute constraints. To that end, we sought to follow these four principles:
Accessibility

One of the important goals of the REIP platform is to provide researchers with a broad range of expertise and backgrounds the capabilities to perform environmental sensing projects. We thus chose the Python programming language for the REIP SDK because of its wide adoption, shallow learning curve, and wide ecosystem of libraries spanning countless domains, including: data science and machine learning. REIP’s API choices also take design inspiration from popular machine learning frameworks that excel at defining connections between different components, which are already familiar to many engineers working with data.

Extensibility

In order to address the requirements of a diverse set of applications, we designed the REIP SDK in a modular fashion. The atomic component of the framework is a Block, which represents one computational unit (e.g., acquiring an image from a camera or applying an object detection model, etc.) with a variable number of inputs and outputs.

Multimodality

The REIP SDK was designed to make minimal assumptions about the data that is being processed, allowing it to be used in a multitude of contexts. The primary constraint is that the data be serialisable for cases where inter-process communication is required. Any domain-specific implementation details are delegated to custom block implementations, which promotes a clean and principled separation of concerns between data engineering and the research problem domain.

Scalability

Python, like many programming languages, provides options for scaling code to run multiple operations at the same time through concurrent programming, commonly known as multi-threading and multi-processing. REIP takes advantage of this by executing each block in its own thread, allowing them to run independently to minimise the latency and maximise the data throughput. The bottleneck in multi-process Python applications is often data serialisation, and we leverage Plasma Store [44] for an efficient shared memory implementation where child blocks can access the read-only version of the data with fixed memory mapping overhead.

### 3.4. Programming Interface

The REIP SDK takes much of its API inspiration from graph definition in Keras [46] and Scanner [47], and its usage consists of two stages. The first is a computational graph/pipeline definition stage. Here, the user declares all of the blocks that are going to be used in the pipeline, how they are being distributed in a multiprocessing context, and their inter-connections, so that they are able to pass data from one to another. An example of this is shown in Figure 2b. Note that none of the data processing code is being executed at this stage.

Once the graph is defined, finally, we can execute it. This is done by simply calling graph.run(), which will spawn all of the graph’s children and begin data processing. The behaviour of this stage is all controlled within the block class definition. We elaborate on each of the SDK components in the following sections.

#### 3.4.1. Blocks

A Block is a fundamental component of the REIP SDK, and is implemented as a Python class that represents one unit of computation (e.g., get audio from the microphone, compute machine learning outputs, upload data to server etc.). Each Block runs in its own independent thread and uses Queues to pass data to others.

Blocks are designed to be easily extendable to suit diverse use cases. A Block consists of: an initialisation function init() that is called at the start and which can be used to acquire resources and set initial values; a process function process(…) that is called repeatedly with data from parent blocks as an input and returns 0 or more outputs to the next block(s); and a cleanup function finish() that is called at the end to release any resources acquired. This general program structure encapsulates a wide family of programs and is fundamental in Object Oriented Programming and Python context managers. An example block implementation is shown on Figure 2a. Note that for a custom block, any of these functions can be omitted if not used.

Blocks can operate in four approximate roles describing how they relate to the data that they are handling (definitions are not binding):Data source (0 inputs, ≥1 outputs, e.g., sensing device such as a microphone);Data processing (≥1 inputs, ≥1 outputs, e.g., object detection in an image);Data sink (≥1 inputs, 0 outputs, e.g., data storage to disk);Operational (0 inputs, 0 outputs, e.g., disk usage monitoring).

Pipelines can be constructed by connecting a data source block to any number of data processing and/or sink blocks. Operational blocks are typically considered as standalone blocks that perform operations without needing to communicate with any other blocks, e.g., a block that monitors and maintains the network connectivity or available disk space.

Users are also able to customise the rules around when a block will execute. For example, a user can customise the strategy used to determine when the framework should call the process function based on the input queue status. So one can change whether a block needs to wait for all inputs to have a value or if it should be executed when at least one of them has a value. A user can also configure the max rate at which a block will run, or the strategy used to get items from the queue, e.g., should a block process the latest value in the queue only or process every buffer.

In its current state, the REIP SDK offers dozens of blocks covering audio tasks (recording, SPL computation, etc.), video tasks (recording, pose, object or motion detection), data output, data encryption, data upload and general utilities. They are organised into corresponding block libraries that serve two main purposes. The first is to speed up the development of sensing applications by means of reusing pre-existing blocks for common tasks. The second, more subtle, benefit of having libraries of blocks that follow a standardised design pattern is documentation of how to perform various tasks in the sensor network building context. We believe that community contributions will greatly extend the range of supported sensing modalities and operations that can be executed on acquired data.

#### 3.4.2. Graphs

Any interesting application will consist of multiple blocks connected together. In order to control multiple blocks at once, they can be assigned to a Graph, allowing them to be spawned, joined and managed together. This joint management of blocks also allows the framework to coordinate when one block experiences errors, the others can either continue running, pause or shut down.

Adding blocks to a graph is very easy (see Figure 2b) and involves simply defining the block inside of the graph’s context (i.e., define them indented under the with statement).

#### 3.4.3. Tasks

By using Blocks and Graphs, we are able to define a data processing pipeline that utilises multi-threading in a single process. However, a single Python process is constrained by Python’s Global Interpreter Lock (GIL), which allows for the parallelisation of the IO-bound code, but not CPU-bound code. In order to utilise all of the available CPUs efficiently, we need to use multi-processing to spawn multiple Python processes (with independent GILs) that can distribute the blocks to run on all of the CPUs. For this, the REIP SDK provides a special type of a Graph, called Task, that works much like a Graph except that all blocks added to it will be executed in a subprocess controlled by that Task object. They have an identical usage as can be observed in Figure 2b.

Blocks are able to detect when their connections are spanning different tasks, meaning that the end user does not need to worry about the logistics of passing data between processes. Users have the ability to specify the type of serialisation to use that best suits their data type and volume, which provides flexibility and the opportunity for optimisation. Communication with Apache Arrow Plasma Store [44] is built into REIP’s cross-task data passing and can be enabled by passing *throughput = “large”* when defining a connection between blocks. It provides efficient, high-volume data throughput where required (Figure 2b). Other serialisation options include the standard Python Pickle method (low throughput) and Apache Arrow’s default serialiser (medium throughput).

Error handling is another problem for multiprocessing and multithreading and, by default, it is difficult to report errors back to the main process/thread. The REIP SDK handles this within Blocks and Tasks for the user and will raise unhandled Block and Task exceptions in the main thread/process.

#### 3.4.4. Data Formats

If a user has a specific sensing problem, they can easily extend the REIP SDK by following the I/O specifications between blocks. Input and output buffers consist of an arbitrary data payload and a dictionary with metadata:buffer = (data, metadata)

Data is the primary data payload, and is commonly (though not necessarily) a Numpy array. By convention, if an output array contains a temporal dimension, then it should appear first and channel information should appear last. For example, video clips would have the dimensions [Time,Height,Width,Channel] and audio data would have the dimensions [Time,Channel].

#### 3.4.5. User-Defined Blocks

Figure 3a illustrates the process of implementing a data source block in a seismic sensing application (new modality). The user can focus on the interaction with the sensing device (sample readout) when implementing this block and, after the conversion of the data format to comply with the REIP API, can immediately get access to and benefit from the vast REIP SDK infrastructure (Figure 3b). A generic Rebuffer block is used to aggregate the samples into batches that can be processed using a re-purposed STFT (Short-Time Fourier Transform) block from the audio library. The rest of the functionality needed to produce a fully functional sensor, such as data storage to disk and data upload, is also available in the REIP SDK. Additionally, with a single line of code, the SeismicSensor block can be put into the context of a Task to ensure that the computationally intensive STFT block does not interfere with any data readout operations that need to be performed at a high rate. This is achieved by means of execution of the SeismicSensor block in a separate process managed by the task, with data passing between these processes handled transparently by the REIP SDK. All this functionality is achieved with less than 30 lines of code (comments excluded).

#### 3.4.6. Data Security

There can be many concerns around data security when it comes to IoT devices, whether it be about remote access to the devices, interception of data upload, or direct access to data storage cards. Most of these concerns are outside the scope of REIP SDK, as addressing them requires OS-level handling. Many can be circumvented though by thoughtful system design, such as protecting outside connections to the devices using firewalls, SSH keys, VPNs for secure remote access and using HTTPS for data upload.

A harder to remedy issue around IoT (or any unaccompanied computer system for that matter) is that a hard drive is non-trivial to secure. For IoT devices, they often need the ability to reboot themselves in the case of system failure or power interruptions. This poses problems when trying to fully encrypt the hard drives, because a password login would be required whenever there was a reboot. The other option is physical security, i.e., making the SD card more difficult to access; however, it also makes it more difficult, and potentially costly, to repair the sensors. Therefore, encryption of sensitive data using a two-sided encryption key is important to ensure that the data cannot be directly accessed from the hard drive itself and can only be decrypted by the main server where the decryption key is stored securely. REIP SDK provides blocks for this type of encryption, including a two-stage encryption technique that reduces network bandwidth for decryption on the server by only needing to transmit a small payload instead of the full data.

## 4. Evaluation

In this section, we implement a number of benchmarks to evaluate the performance of the REIP SDK in different application scenarios. First, we implement a video processing pipeline to understand the performance of the concurrency tools offered by the REIP SDK (Blocks, Tasks, etc.) under different configurations (Section 4.1). We also take a closer look at the serialisation strategies available (Section 4.1.2), and in which configurations they are the most beneficial, as well as compare the performance of the REIP SDK to existing frameworks (Section 4.1.5). An audio processing pipeline is then evaluated on different hardware platforms (Section 4.2) to estimate the performance overhead of the REIP SDK and to identify the optimal hardware platform (Section 4.2.2) for building a physical sensor prototype powered by the REIP that will be used in our case study (Section 5).

### 4.1. Concurrency

One of the typical problems that users face when building sensors is processing data in real-time. When using Python to create a data processing pipeline, this will inevitably involve multi-processing and inter-process communication (Section 3.4.3), where serialisation to pass objects from one process to another can be incredibly resource intensive, particularly for large data arrays. With the REIP SDK’s concurrency tools, such as Tasks, the user can easily implement a pipeline that meets their project’s requirements. The focus of the following subsection is on the evaluation of these concurrency tools under different pipeline configurations.

#### 4.1.1. Test Pipeline

We evaluate the concurrency tools of the REIP SDK using a video processing pipeline (Figure 4a) consisting of data acquisition with two camera blocks (5 MP, 14.4 fps) followed by real-time object and motion detection blocks, and the corresponding image writer block.

Different variations of the video pipeline were executed on the NVIDIA Jetson Xavier NX platform (up to 6 CPUs with 8 GB of RAM and 384 CUDA cores) for 30 s, and the number of frames processed by each block were measured (Table 2). Object and motion detection blocks use the ‘latest’ connection strategy, while image writer blocks process every frame provided to them by object or motion detection blocks. The motion detection block is outputting one difference image for every two consecutive input frames. The object detection block is next overlaying the bounding boxes of the detected objects on every processed frame and outputs it to the image writer block. We also report the number of lost frames that have not been pulled by the camera block in time for scenarios were the system gets overloaded, as well as any frames still left in the input queues of the image writer blocks. For stereo configurations, object and motion detection blocks are alternating their input on every call of the process function.

#### 4.1.2. Serialisation

The REIP SDK includes a few serialisation methods for the user to choose from to meet their bandwidth requirements. The first method is the most common and uses Pickle, a package in Python’s standard library that is capable of serialising arbitrary Python objects. Looking at Figure 4c, we can see that Pickle offers the fastest speeds for small buffers (<0.3 MB, throughput = “small” when connecting blocks), but Apache’s Plasma Store [48] (“Plasma” in plot) is fastest for larger buffers (throughput = “large”). The Plasma Store is a shared memory server that uses Apache Arrow to serialise and provide fast, copy-free memory views of the data that are much more efficient for large data arrays (approaching the memory speed limit).

The final serialisation method, Pyarrow, uses the same serialisation method as Plasma Store, but it does not provide shared memory (throughput = “medium”). It has a slight performance increase over Pickle for buffers greater than 0.5 MB in size. See Table 2 for comparison of the impact of different serialisation strategies on the overall performance of the video processing pipeline (REIP Hybrid).

#### 4.1.3. Configurations

The following pipeline configurations are being evaluated:*REIP Hybrid*: image writer blocks are executed in the same task as the corresponding object and motion detection blocks. This is our primary configuration as depicted on Figure 4a.*REIP Multiprocessing*: each block is executed in an independent task (a full multiprocessing configuration).*REIP Threading*: all blocks are being executed in the same (main) process but, by design, in their own independent threads.

We further customise the configurations by using different serialisation strategies (Pickle/Pyarrow/Plasma) for inter-process block connections. The system is also intentionally throttled by enabling only four CPU cores to measure smaller variations in performance overhead under these different configurations. For the most fair comparison with other representative frameworks (Ray and Waggle), we implement a thin wrapper emulating the REIP API for each framework (more details in Section 4.1.5). We use the same wrapper for the REIP SDK too (REIP Backend in Table 2), which does not include different connection strategies or other advanced features of the REIP SDK (hence a slightly better performance compared to the full REIP SDK).

#### 4.1.4. Performance

The performance metrics in this experiment are the number of frames/buffers processed by each block in the pipeline, so the higher the number in *Pulled* (number of frames successfully acquired by a *Camera* block), *Detected* (number of frames processed by an *Object* or *Motion* detection block), and *Saved* (number of images written by the respective *ImageWriter* blocks) columns in Table 2, the better the performance of the pipeline. Conversely, the target values for the *Lost* (number of frames missed by *Camera* blocks due to system overload) and *Queued* (number of frames processed by *Object/Motion* detection blocks but still pending to be saved by the *ImageWriter*) columns are 0 for optimal performance. It should also be noted that the values in different columns are not directly comparable in isolation. The pipeline configuration that processed less of the captured frames but saves more of the processed results can be argued to perform better than the one processing everything but saving little of the results. We provide a complex analysis of the measured metrics in the remaining sections.

It is apparent from Table 2 that all stereo multiprocessing configurations are not capable of processing all camera frames in the throttled scenario, with a clear trend of higher performance for higher throughput serialisation. Nevertheless, the hybrid approach does succeed in this task for single camera video. The threading only configuration shows the highest performance as it does not need to incur any additional performance overhead due to serialisation, with the maximum CPU resources spent processing and saving the images (this does not mean that such a configuration is optimal in a general case). Image writing is largely limited by the maximum disk write speed and the remaining CPU resources available after object/motion detection. Some configurations (e.g., REIP Multiprocessing) are less efficient in data management and result in less of the results being saved (compared to REIP Hybrid) despite high processing rates.

It should be noted that it is only in this specific data acquisition and processing pipeline (Figure 4a), where both cameras are operating at the same frame rate (14.4 fps) and object/motion detection blocks are highly optimised (object detection is performed on GPU), that we do not observe data loss for camera blocks due to the Python GIL in the threading only configuration. The data loss would be inevitable when using data sources with significantly different sampling rates. For instance, the microphone block requires being serviced more often than the camera block and will experience overrun errors with data loss due to the Python GIL if there is another block (in the same task/process) performing heavy computation that does not release the GIL in a timely manner. This makes the Task feature of the REIP SDK essential for easy decoupling of data source blocks from data processing blocks by placing them into a dedicated task/process with an independent Python interpreter.

#### 4.1.5. Comparison with Other Frameworks

To compare the REIP SDK with other existing frameworks, such as Ray or Waggle, we implemented a thin wrapper layer to interface the blocks’ implementation with different execution backends (Table 2). For REIP, we are using a high throughput hybrid configuration of the video processing pipeline (Figure 4a). For Waggle, we map each task in the pipeline to a Waggle plugin and since the Waggle framework does not provide concurrency tools (only a RabbitMQ wrapper for data upload), we use a standard multiprocessing queue to transfer frames between different plugins. In the Ray backend, we use ray.remote futures for parallel processing.

Table 2 shows that the performance of the REIP backend is higher than the large throughput hybrid configuration executed with the REIP runtime, closer to REIP Multiprocessing. This is expected because the wrapper layer does not provide all of the features of the REIP SDK (e.g., detailed statistics and error handling), which introduce extra overhead.

In turn, the performance of the Waggle backend is similar to the REIP Hybrid configuration with small throughput because the same serialisation method (Pickle) is being used. It is a bit lower though (stereo configuration in particular) because of a lack of multithreading in the object and motion detection plugins/tasks.

Finally, Ray provides great concurrency tools but they were developed with different design constraints in mind, which is reflected in its performance. Because Ray uses futures and lazy/deferred evaluation, we observe an execution pattern, where a number of jobs are being accumulated and then executed as a batch that effectively halves its performance. We were also not able to get a GPU accelerated object detection block working with the Ray backend because the Ray framework does not recognise the Jetson’s GPU.

### 4.2. Overhead

Another important aspect of any software framework is its performance overhead. Scalability is one of the design principles of the REIP SDK and for that, its internal routines have been optimised to minimise the amount of service time and maximise the computational resources available for the execution of user code. The percentage of time spent by the framework performing data management and other service routines also depends on the particular computing platform in use, which can vary in the amount of RAM available, number of CPU cores and their speed. The focus of this subsection will be on the performance overhead of the REIP SDK on different hardware platforms.

#### 4.2.1. Test Pipeline

We have implemented an audio processing pipeline (Figure 4b) that resembles a real-world noise pollution monitoring sensor network (The Sounds Of New York City project or SONYC [49]), excluding the data encryption and upload functionality (bottom-left blocks with dashed lines in Figure 4b). In this setup, a mono microphone is recording one second long audio snippets that are supplied to the sound classification and sound pressure level (SPL) computation blocks for real-time audio analysis. The results are saved to CSV files alongside the raw audio data in 10 s intervals after rebuffering. The most computationally demanding is the sound classification block, where we use a CPU-only implementation of a neural network-based classification model to make the test pipeline compatible with a wider range of computing platforms that may not have GPU support. We have also placed the microphone block in its dedicated audio capture Task to avoid any overrun errors, as explained in Section 4.1.4.

#### 4.2.2. Hardware Platforms

The REIP SDK is Unix-platform compatible so it can be installed and executed on a large variety of single board computers (SBCs) at various price and power points. This flexibility allows the software to be matched with suitable hardware based on the computational demands of the blocks used in the pipeline and the project budget. We explore the performance of a real-time, high-resolution audio processing pipeline (Figure 4b) on different multi-core CPU SBCs with at least 512 MB of RAM (Table 3).

#### 4.2.3. Performance

The key performance metrics are the percentage of time spent by each block performing data processing, awaiting the next buffer and the service time of the framework managing the data delivery between different blocks. We also record the number of dropped buffers in the sink queues of each block output to identify performance bottlenecks in the pipeline (if any). The criteria for selecting the computing platform is its ability to execute all of the included processing blocks at greater than real-time.

Results in Table 3 show that even a low-end embedded platform, such as the Raspberry Pi 4B, is capable of processing all the data in real time using the REIP SDK without dropping a single buffer. The service time overhead of REIP SDK remains well under 10% on all platforms, which is negligible.

### 4.3. Discussion

Real time data capture and edge processing present many challenges in terms of managing compute performance and overall data transfer on device. Existing software frameworks in this domain are often designed with assumptions that data transfer overhead, as a result of serialisation, is negligible compared to the anticipated computational load or are designed to be used on large compute clusters without the computational or concurrency constraints inherent in lower power edge sensors. In this section, we have highlighted that the REIP SDK can handle complex data processing pipelines that outperform other approaches, such as Python’s multiprocessing or other frameworks (e.g., Apache Ray).

An important and often overlooked aspect of data processing pipelines is the moving of data between processing blocks. Serialisation of data so that it can be exchanged between these blocks can impose significant overhead to complex pipelines, especially when these data are large in size. REIP’s hybrid approach to serialisation shows comparable or better performance than other frameworks in the complex video processing pipelines presented. The ability to define a “small”, “medium”, and “large” throughput flags (defined in Section 4.1.2) when establishing block connections provides a way to select the optimal serialisation technique applied for the scale of data being passed between these blocks.

When implementing an audio-based processing pipeline on various embedded compute platforms typically used in sensor networks, service times stayed below 10% when using the REIP SDK. The consistency of these service times is worth noting, as they stay relatively steady between hardware platforms, which suggests that the serialisation approach of the REIP SDK is agnostic of the platform it is running on. These service times are likely driven mainly by the memory architecture of the device which impacts serialisation, rather than its computational power. We consider the service overhead of the REIP SDK in these examples to have a minimal impact on overall performance and generalise well to different hardware platforms; however, further work is needed to compare the service times of other frameworks.

In REIP, we are targeting the design, and more specifically in this paper, the code implementation and computational challenges that often arise during the design, development and deployment of individual sensors as well as sensor networks. The REIP SDK aims to provide to the user the tools necessary for efficient utilisation of limited computing resources, while remaining easy to use and flexible in development.

## 5. Case Study

To showcase this initial implementation of the REIP SDK, we present a real-world research case study using a custom multimodal sensor design for real-time traffic analytics with a focus on bicycle accidents. This sensor network is designed for long-term deployment on light poles at busy urban intersections, where power is available but high speed Ethernet or Wi-Fi-based internet connectivity is not. Researchers want to understand more about the frequency of bicycles that pass through the intersection but also want to analyse video and multi-channel audio of any close interactions between bicycles and other vehicles on the road to study the circumstances around bicycle accidents and the use of multimodal sensor systems to detect these events.

This section focuses on the development workflow steps when using the REIP SDK as described in Section 3.1. In this workflow case study, steps 3 and 5 have already been addressed through the evaluation of the processing pipeline with hardware selection and the installation of all the necessary runtime software. Due to budget constraints, we have chosen the cheapest of NVIDIA’s SBC range with GPU support which, due to the computational requirements of the presented case study, limits us to using only one of the two cameras available in the sensor prototype. Since it is also sufficient for this case study to use one sensor prototype instance, step 5 in the REIP workflow (Figure 1) can be skipped, but that by no means limits the number of sensors that can be used in a sensor network powered by the REIP SDK. The more thorough evaluation of sensor hardware and the experience of researchers is a subject of future studies.

### 5.1. Definition of Requirements

For this use case, the following functional requirements must be satisfied by the application pipeline:Video capture with quality higher than 720p at 15 fps;Multichannel audio capture;Object tracking of vehicles;Transmission of traffic analytics and raw multichannel audio and video of accident near misses via limited cellular plan.

A major constraining factor in this application is the reliance on data limited cellular connectivity. This excludes the possibility of retrieving continuous video and multichannel audio data for the post hoc detection of accident events, which would exhaust cellular plan data limits of ≈50 GB/month very quickly. To alleviate the impacts of this constraint, we can leverage recent advancements in compact but high-power compute devices to push the event detection processes to the sensor itself, so only salient events are transmitted over constrained network. This is a common need in longitudinal sensing research, where in practice the large majority of collected raw data contains few or no events of interest.

### 5.2. Implementation of Application Pipeline

To highlight the flexibility of the REIP SDK, we have created an application pipeline that constitutes a multimodal *smart traffic event detector* for urban bicycle accident data collection. Figure 5 shows the block diagram of this pipeline, where raw video and multichannel audio uploads are preserved only when a salient event is detected, which in this case is the near miss of a cyclist with a motorised vehicle. To ensure that the maximum amount of useful data is retrieved, both video and audio cues are used to signify a salient event. For video, object detection block using the MobileNet V2 [50] model is fed with video frames at 4 fps from the GStreamer [29]-based camera block, which is also recording the full speed (≈15 fps) video feed into files. An event detector block independently monitors the output of the object detections and deletes the corresponding video fragments if a bicycle and motorised vehicle were not captured in the same frame for any frame in the video fragment. In cases where the camera and object detector block struggle to capture an event, such as: under low-light conditions, when an event is off-camera, or when it is occluded, the SPL (Sound Pressure Level) computation block will report a spike in decibel level, caused, for example by a horn blast, impact, or tire screech from emergency braking. This information is also used by the event detection block to decide on which data fragments contain events of interest and should be preserved.

### 5.3. Hardware and Software Integration

Using the block design, each data source block has a corresponding physical sensing device (e.g., a camera or a microphone) that connects to the computing platform. Here, we define our customised sensing devices (that are compatible with REIP’s default data source blocks), and implement any custom blocks required. The prototypical sensor system shown in Figure 6 uses two 5 MP USB cameras providing a 160° horizontal field of view (85° max per camera) and 15 fps recording, satisfying our video capture use case requirement. The compute core is the NVIDIA Jetson Nano Developer Kit [5], which offers edge intelligence capabilities from its 128-core GPU, quad-core 1.43 GHz CPU and 4 GB of LPDDR4 RAM. The majority of the sensor’s hardware is enclosed within an aluminum weatherproof housing.

The custom acoustic front-end has been designed to capture synchronised 12 channel audio from its 4 × 3 array of digital pulse density modulated (PDM) Micro Electro Mechanical Systems (MEMS) microphones. It uses the USB MCHStreamer [51] as an audio interface, so the same Microphone block from the REIP audio library can be reused to read data from this USB audio class compliant device. In the given application pipeline, a single microphone channel is used as input to the audio processing blocks. For data transmission, a USB LTE modem is integrated containing a SIM card providing a bandwidth limited cellular plan.

### 5.4. Experimental Deployment

A single sensor was mounted on a tripod at 1.8 meters above the ground facing a roadway in a quiet urban parking lot. This location was chosen to allow for sustained data capture in more controlled conditions than a busy urban intersection. The application pipeline was executed prior to multiple runs of vehicle passes, including: (1) motorised vehicle (car, truck or motorcycle) passes with and without horn use, (2) bicycle and pedestrian passes and (3) various combinations of simultaneous passes. A total of ≈1 h of continuous data collection was performed, which would equate to ≈53 GB/h of dual 5 MP video with MJPEG compression and 12 channel audio if all data were to be captured and uploaded. These amounts of data would overwhelm the majority of cellular data plans, requiring at least 15 MB/s of bandwidth to transmit these data from edge sensor to server, regardless of the inevitable saturation of monthly data plan limits.

Figure 7 shows one frame captured from the left video camera for illustration purposes. At this instance, a cyclist was passing by the sensor with two cars driving in the opposite direction while honking their horns. The object detection block has returned confident detections including their bounding boxes around the car, person and bicycle. The event detection block raised an event flag, as there was the combination of a cyclist and motorised vehicle within the frame, which resulted in the successful upload of 10 s of video and multichannel audio of the event via the cellular modem. At the figure bottom, time vs. SPL is plotted with the green region showing the portion of time that the object detector block is able to confidently identify the motorised vehicle passby, mainly due to the limited field of view of the camera setup. Of note is the much wider red region showing the extent over which the SPL computation block is able to detect the elevated sound level of the car horn, highlighting the advantages of this multimodal approach.

### 5.5. Discussion

Practically, this smart traffic event detection sensor dramatically reduces data throughput requirements for research projects looking to analyse urban phenomena. The rapid implementation of this real-world application following REIP’s workflow and using the REIP SDK shows promise for its use across diverse applications of sensor networks. With a given set of application requirements, a pipeline can be designed that abstracts away a large amount of technical detail typically required to develop software that could perform this set of complex, interacting operations.

A particularly challenging aspect of real-time sensing is moving data between processes that operate on different schedules. This is commonly addressed with ad hoc solutions and application-specific code that are not generalisable or re-configurable. When projects are under tight deadlines, it often does not make sense to do more than that; the project just needs code that will perform the required task, and re-usability and portability are not the primary motivators. The REIP SDK addresses these common computational problems in a general sense so that when a research problem comes along, an experiment can be up and running as quickly as possible using pre-existing functional blocks. Indeed, we were able to reuse many of the blocks used during evaluation in Section 4 to build the data acquisition and processing pipeline of the given case study (Figure 5).

Another significant challenge in remote sensing is hardware and software integration, which is subject to a number of constraints including: compute resources available, hardware I/O offered, sensing options, inter-process data rates and available remote connectivity options. With an application pipeline defined using the REIP SDK, this integration process becomes less of a challenge, as the blocks chosen dictate the minimal hardware platform that can support it. The presented process of manually benchmarking possible hardware platforms (Section 4.2.2) is not ideal but is a precursor to our planned simulation and optimisation tool for a pipeline evaluation stage, where optimal hardware platforms will be matched to an application pipeline in an automatic way subject to user constraints, such as maximum memory usage, data output rate, etc.

## 6. Conclusions and Future Work

The case study in Section 5 highlights the use of the REIP SDK in the design of an application pipeline for a multimodal smart filter for urban bicycle accident data collection. A finding from this case study was that the REIP workflow (see Section 3) allows the sensor architect to design the data pipeline from an abstract, top-down view and have it translate directly into software components without the extraneous complications caused by software and hardware-specific details. The modular blocks that were combined to make up the case study’s functionality abstracted a lot of the complexity away from the traditional software development process. Moreover, it allows hardware decisions to be pushed to a later stage in the process which, if made too early, can unnecessarily constrain the application.

The REIP SDK includes dozens of blocks for the commonly required tasks of: data acquisition, processing and storage. Advanced data serialisation and block connection strategies (Section 4.1.2 and Section 3.4.3) offered by the REIP SDK can then be used to quickly find the right balance between multi-threading and multi-processing for optimal performance, which is crucial for real-time data processing at high rates with minimal loss of samples. It is also important to note that the REIP SDK can handle data of varying structure and size, and does not add any significant overhead (Section 4.2.2) so the user application can maximise the utilisation of available computational resources.

To further increase the usability and accessibility of the REIP platform, we are currently working on a Graphical User Interface (GUI) for application pipeline creation using the Node-RED [52] platform. This browser-based interface will provide a visual representation of the functional blocks. It will be used to wire-up blocks and create application pipelines for less technical users. This GUI can take into account the constraints of the proposed system, as well as allow users to define custom application-specific hardware/software constraints. The resulting application pipeline can then be exported in the form of a deployable script that is ready to be executed on sensor nodes with our run-time installed. In addition, for embedded devices such as the Raspberry Pi and NVIDIA Jetson, the compilation steps for dependency libraries can be long, difficult and error prone. In order to ease this pain (for ourselves and others) we also plan to provide working Docker images that already contain the core library dependencies of the REIP SDK, such as PyArrow, and others that are typically difficult/time-consuming to build on resource constrained devices.

The REIP SDK was designed to facilitate and streamline the process of environmental sensing deployments, providing researchers of varying experience levels with tools and best practices for designing and building sensor networks (Section 3.2). Implementing sensor networks can end up being an enormous engineering burden and often takes away valuable time from the actual research that the researchers are seeking to perform. We hope that the REIP SDK presented can alleviate this burden.

## 7. Patents

The work resulted in a utility patent application number US20210287336A1.

## Figures and Tables

**Figure 1 sensors-22-03809-f001:**
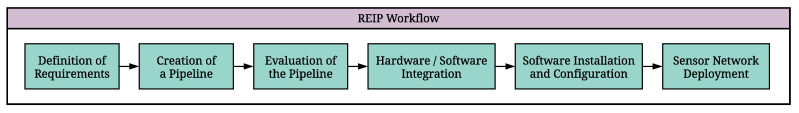
Typical steps of the sensor network prototyping process using REIP.

**Figure 2 sensors-22-03809-f002:**
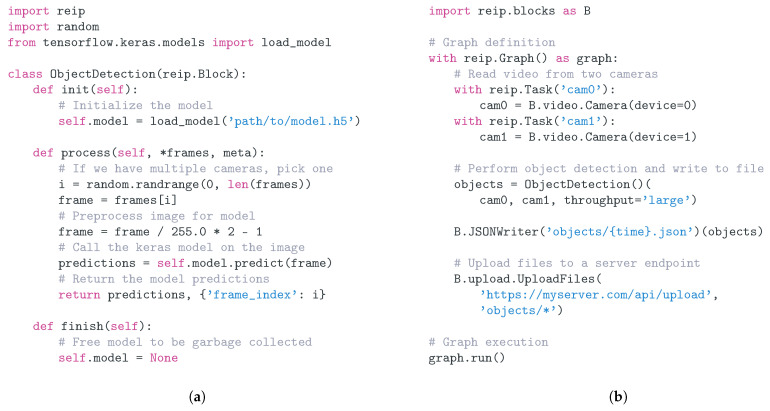
Code examples illustrating the usage and extensibility of REIP SDK in video capture and processing application. (**a**) Block implementation for a machine learning model. It consists of an init, process(…) and finish methods. The process method takes frames from multiple cameras and chooses one to perform object detection on it. (**b**) Graph definition showing object detection on a video stream with the detections saved in JSON files and uploaded to an API endpoint.

**Figure 3 sensors-22-03809-f003:**
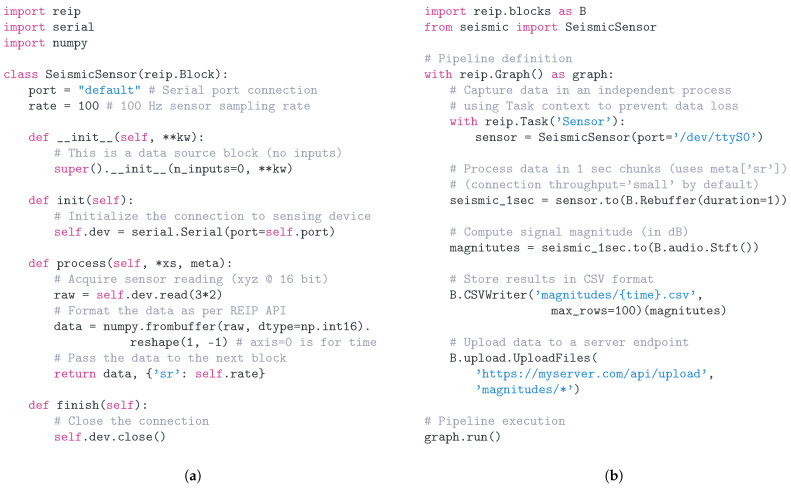
Code example illustrating how to extend REIP SDK to work with new sensing modality. (**a**) Block code for capturing seismic sensor readings, formatting them according to REIP API and passing the data processing pipeline downwards. (**b**) Graph definition code demonstrating how to capture a new modality while reusing existing REIP blocks (e.g., Short-Time Fourier Transform (STFT)).

**Figure 4 sensors-22-03809-f004:**
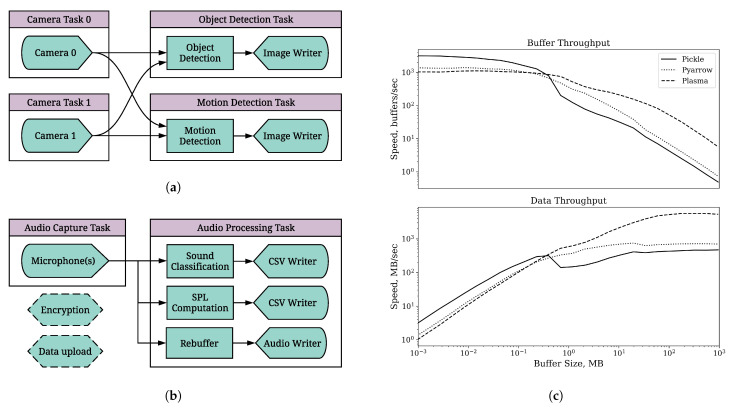
Data processing pipelines used to benchmark REIP SDK (left) and comparison of different serialisation strategies. (**a**) Video processing pipeline for investigation of REIP SDK performance overhead in different configurations (Table 2). (**b**) Audio processing pipeline for comparison of the overall performance of REIP SDK on different platforms (Table 3). (**c**) Data throughput between two blocks as a function of buffer size and serialisation strategy for inter-process communication on Jetson Xavier NX. Pickle serialisation has the lowest overhead for small buffers whilst Plasma method provides the highest data throughput for larger buffers.

**Figure 5 sensors-22-03809-f005:**
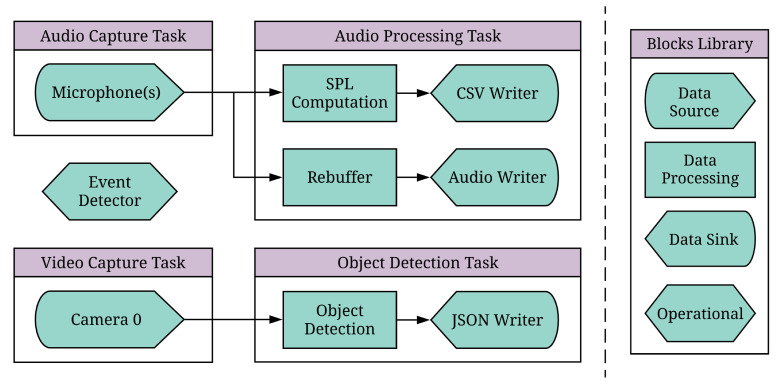
Multimodal smart traffic event detection pipeline. This pipeline handles both audio and video. It captures video, passes it to an object detection Block, and then writes the detection outputs to JSON files. It also captures audio at short intervals (1 s) and computes SPL at the 1 s resolution, as well as accumulates the audio into longer clips and writes them to disk as audio files. Finally, the salient event detection block is monitoring these files, preserving only those that had an event of interest detected in them.

**Figure 6 sensors-22-03809-f006:**
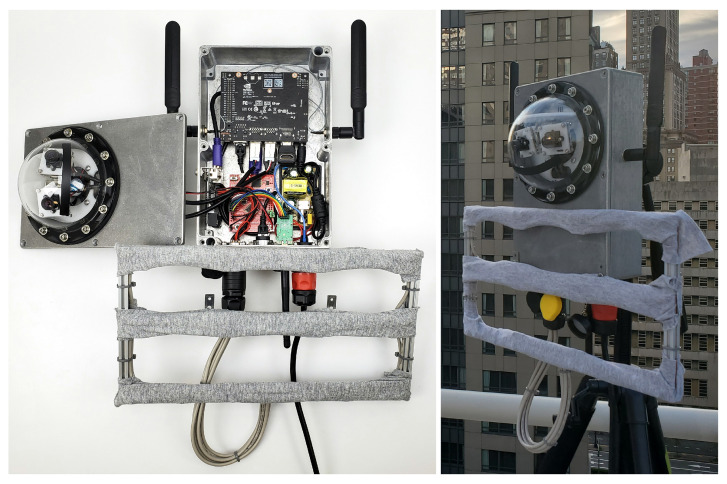
Multimodal sensor built for the real-world case study. It includes two 5 MP cameras, a 12 channel microphone array (4 × 3) and a cellular modem. The sensor is waterproof for outdoor operation and powered by the NVIDIA Jetson Nano, which is thermally coupled to the aluminum enclosure for passive/silent cooling.

**Figure 7 sensors-22-03809-f007:**
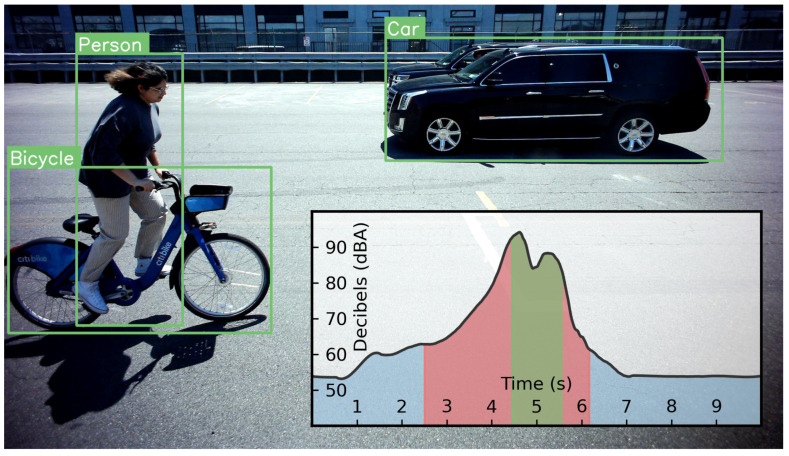
Car and bicycle pass-by frame from sensor’s camera overlaid with bounding boxes and labels from object detection block for illustration. Green region on inset audio amplitude plot shows time period where car is detectable within video frames, with red region showing longer periods of audio-based detection of a car horn. The blue portions signify periods when no audio event was detected.

**Table 1 sensors-22-03809-t001:** Related works feature comparison. The features include (from left to right): open source availability; user-friendly API/low barrier for entry; support of multiple sensing modalities; ability to easily add new features or sensing modalities (extensible); ability to handle large amounts of data (scalable); sensor level integration (HW/SW); and whether the framework can be used with various computing platforms and sensor devices (device-agnostic). The ✓ and × symbols denote whether the feature is present or not in each framework. The parenthesis indicate that the feature is present in the given framework but it does not excel at it, or that the framework could possibly be used but it was not designed to have such feature. A question mark means no assessment.

Project Name	Open-Source	Simple API	Multimodal	Extensible	Scalable	HW/SW	Device-Agnostic
FIT-IoT Lab [12]	✓	✓	✓	×	✓	✓	×
FIESTA IoT [13]	×	×	✓	×	✓	✓	✓
Signpost [14]	✓	(✓)	✓	✓	×	✓	×
SensorCentral [15]	×	?	✓	✓	✓	×	✓
Array of Things [16,17]	✓	(✓)	✓	✓	×	✓	×
WaspMote [18]	✓	✓	✓	(✓)	×	✓	×
The USC [19]	×	×	?	?	✓	✓	?
FIWARE [20]	✓	(✓)	✓	(✓)	×	×	✓
DDFlow [21]	✓	(✓)	(✓)	(✓)	(✓)	×	✓
EdgeProg [22]	×	?	×	×	(✓)	(✓)	(✓)
Caesar [23]	×	?	×	×	?	×	(✓)
Waggle [24,25]	✓	✓	✓	✓	×	✓	(✓)
Apache Ray [26]	✓	✓	(✓)	(✓)	(✓)	×	(✓)
Celery [27]	✓	✓	(✓)	✓	×	×	✓
Spotify Luigi [28]	✓	✓	×	×	×	×	(✓)
GStreamer [29]	✓	×	(✓)	(✓)	✓	×	(✓)
NVIDIA DeepStream [30]	✓	×	✓	(✓)	✓	(✓)	×
FFmpeg [31]	✓	×	(✓)	×	(✓)	×	(✓)
REIP (SDK)	✓	✓	✓	✓	✓	✓	✓

**Table 2 sensors-22-03809-t002:** Performance of different framework configurations when running a video processing pipeline (Figure 4a) on Jetson Xavier NX. Values denote number of frames processed by different blocks in the pipeline during 30 s sampling period. Top half of the table corresponds to the mono and bottom half to the stereo camera configuration. The system was throttled to use 4 CPU cores only to be able to measure more subtle performance differences between different configurations. Negative queued values indicate extra frames processed that have already been in the queue prior to the sampling interval. Symbol ⌃ indicates queue overflow (max queue size was set to 100 buffers).

Configuration		Camera 0		Camera 1		Object Detection			Motion Detection		
Name	Serialisation	Pulled	Lost	Pulled	Lost	Detected	Saved	Queued	Detected	Saved	Queued
REIP Hybrid	Pickle	433	0	-	-	206	206	0	191	95	0
REIP Hybrid	Pyarrow	431	0	-	-	229	229	0	196	98	0
REIP Hybrid	Plasma	431	0	-	-	351	204	54 ⌃	160	80	0
REIP Multiprocessing	Pickle	314	118	-	-	101	73	28	104	52	0
REIP Multiprocessing	Plasma	402	30	-	-	385	184	45 ⌃	142	71	0
REIP Multithreading	-	432	0	-	-	388	293	56 ⌃	193	96	1
REIP Backend (Mono)	Plasma	431	0	-	-	420	318	34 ⌃	422	212	0
Waggle Backend (Mono)	Pickle	410	22	-	-	224	132	52 ⌃	202	113	−12
Ray Backend (Mono)	Plasma	432	0	-	-	-	-	-	146	73	0
REIP Hybrid	Pickle	343	89	346	86	126	133	−7	115	57	0
REIP Hybrid	Pyarrow	409	23	321	111	123	122	1	134	66	0
REIP Hybrid	Plasma	327	105	375	57	294	135	39 ⌃	181	90	1
REIP Multiprocessing	Pickle	268	164	325	107	79	66	13	82	41	1
REIP Multiprocessing	Plasma	306	126	299	133	255	121	60 ⌃	208	104	0
REIP Multithreading	-	432	0	432	0	401	263	50 ⌃	346	173	0
REIP Backend (Stereo)	Plasma	433	0	433	0	376	230	64 ⌃	318	159	0
Waggle Backend (Stereo)	Pickle	374	58	356	76	102	86	16	64	32	0
Ray Backend (Stereo)	Plasma	432	0	432	0	-	-	-	149	75	−2

**Table 3 sensors-22-03809-t003:** Time spent by different blocks in the audio processing pipeline (Figure 4b) performing data processing, idle waiting, and servicing the data between blocks. Comparison is given for different embedded platforms ranging from a low-budget Raspberry-Pi 4B to the high-performance NVIDIA Jetson AGX Xavier. The service times remain well under 10% for each platform, which indicates negligible performance overhead introduced by the REIP SDK. No dropped buffers were detected.

Block	Raspberry Pi 4B			Jetson Nano			Jetson TX2			Jetson AGX Xavier		
	Process	Wait	Service	Process	Wait	Service	Process	Wait	Service	Process	Wait	Service
Microphone	0.39%	92.9%	6.68%	0.19%	92.4%	7.28%	0.10%	93.1%	6.72%	0.19%	92.2%	7.33%
Machine Learning	42.1%	50.4%	5.12%	30.0%	61.7%	7.73%	37.7%	56.4%	5.31%	12.2%	78.3%	8.75%
SPL Computation	4.74%	88.5%	6.75%	2.58%	90.0%	7.36%	2.01%	91.1%	6.84%	2.40%	90.0%	7.53%
Audio Writer	1.82%	91.6%	6.58%	0.41%	92.4%	7.20%	0.09%	93.3%	6.63%	0.14%	92.6%	7.25%

## Data Availability

Use case dataset discussed in Section 5 can be accessed at the following location: https://doi.org/10.5281/zenodo.6539485, updated on 5 May 2022.

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
