# Peer review of "REIP: A Reconfigurable Environmental Intelligence Platform and Software Framework for Fast Sensor Network Prototyping"

_sensors, 2022, doi:10.3390/s22103809_

Round 1

Reviewer 1 Report

The paper discusses the design and development of a reconfigurable environmental intelligence platform for fast sensor network prototyping. It is a very interesting read and the contribution is substantial as the prototype offers scalability and fast prototyping, while the presentation of the work is excellent with interesting experimental results. However, there are two issues that could be discussed and further explained in the manuscript.

To begin with, it is well-known that security and privacy are of utmost importance in the IoT world. However, there is not any specific mention in the paper. Especially considering that the solution is presented as a stand-alone product, the authors should try to explain how they handle these two sensitive areas.

Furthermore, there are many platforms that offer similar characteristics and offer their own SDKs. Nevertheless, there is not any substantial comparison to understand how the proposed system outperforms the state-of-the-art. The authors could try to showcase explicitly the benefits of their system.

Other than these two issues, the paper seems really good.

Author Response

The authors would like to thank the reviewer for their feedback which helped improve the manuscript. Please see attached a point-by-point explanation of the applied changes. The manuscript has also undergone an additional proofreading by native English speaker.

Reviewer 2 Report

In this paper, the authors present a fast sensor network prototyping platform titled REIP. The paper presents an interesting software package that can help facilitate research projects in the area.

In general, the current version of the manuscript is at a better position than the first version. However, I would like to suggest the following revisions:

1. The introduction needs further expansion on the context of the problem. The information given does not present the problem definition clearly.
2. It would be helpful to have a summary table at the end of the "Related Works" section to compare the features of these previous works. This would help demonstrate how necessary the proposed system is.
3. Another performance comparison would be useful at the discussions section to help demonstrate the superiority of the proposed system in comparison to previous related works.

Author Response

(The authors gave the same response as above.)
